# Bacterial capture efficiency in fluid bloodstream improved by bendable nanowires

Lizhi Liu[1,2,3], Sheng Chen[4], Zhenjie Xue[1,2], Zhen Zhang[1,2], Xuezhi Qiao[1,2], Zongxiu Nie[1,2], Dong Han[5], Jianlong Wang[3] & Tie Wang [1,2]

Bacterial infectious diseases, such as sepsis, can lead to impaired function in the lungs, kidneys, and other vital organs. Although established technologies have been designed for the extracorporeal removal of bacteria, a high flow velocity of the true bloodstream might result in low capture efficiency and prevent the realization of their full clinical potential. Here, we develop a dialyzer made by three-dimensional carbon foam pre-grafted with nanowires to isolate bacteria from unprocessed blood. The tip region of polycrystalline nanowires is bent readily to form three-dimensional nanoclaws when dragged by the molecular force of ligand-receptor, because of a decreasing Young's moduli from the bottom to the tip. The bacterial capture efficiency was improved from ~10% on carbon foam and ~40% on unbendable single-crystalline nanowires/carbon foam to 97% on bendable polycrystalline nanowires/carbon foam in a fluid bloodstream of 10 cm s$^{-1}$ velocity.

[1] Beijing National Laboratory for Molecular Sciences, Key Laboratory of Analytical Chemistry for Living Biosystems, Institute of Chemistry, Chinese Academy of Sciences, 100190 Beijing, China. [2] University of Chinese Academy of Sciences, 100049 Beijing, China. [3] College of Food Science and Engineering, Northwest A&F University, Yangling, 712100 Shaanxi, China. [4] Department of Mechanical Engineering, Michigan State University, East Lansing, MI 48824, USA. [5] National Center for Nanoscience and Technology, 100190 Beijing, China. Correspondence and requests for materials should be addressed to T.W. (email: wangtie@iccas.ac.cn)

In general, pathogens in the blood of healthy people can trigger serious infectious diseases, such as sepsis, a potentially fatal systemic illness characterized by whole-body inflammation in response to microbial invasion[1, 2]. The general therapeutic strategy for sepsis is to use empiric, broad-spectrum antibiotic therapy[3]. However, these broad-spectrum antibiotics are less effective than therapeutics designed to target specific microbes or antibiotic-resistant pathogens. It has been reported that the blood pathogen load is the main reason for disease severity and mortality in patients with sepsis, therefore extracorporeal blood-cleansing therapy is considered a potential option for addressing the root of the problem, by which the blood pathogen can be quickly cleared without the prerequisite of identifying the infection source or changing the blood contents.

To achieve this goal of cleansing the blood, many methods for isolating pathogens have been examined, such as filtration[4], microfluidic devices[5–8], and magnetic nanoparticle separation[9]. Additionally, combining synergistic effects of surface chemistry (specific ligand/receptor recognition) and nanotopography (suitable micro- or nano-topographical interactions), three-dimensional (3D) nanostructures, including stiff inorganic materials such as Si nanowires (NWs)[10, 11], $TiO_2$ nanosisal-like[12] and quartz NW arrays[13] and softer materials such as polystyrene nanotubes[14], polypyrrole NWs[15], and poly(dimethylsiloxane) (PDMS) microposts[16], have been investigated to capture circulating tumor cells[17] or bacteria[18]. However, these may fail to display an impressive performance in patient's bloodstream conditions because of the bacteria falling off, driven by the shearing force of fluid bloodstream. The common characteristic of these existing isolation technologies is that they introduce relatively weak interaction forces of synthetic small molecules[19], natural lectin[20], and antibodies[21].

Inspired by the natural trapping process of the Venus flytrap, whereby its two lobes open widely and snap shut when stimulated by prey to prevent them from escaping, we describe here an approach to design 3D nanoclaws, improving bacterial capture efficiency in a patient's bloodstream. The 3D nanoclaws are made by bendable polycrystalline NWs pre-grafted on 3D carbon foam (CF). Specifically, in the unclosed state, the bendable polycrystalline NWs are straight (pointing outward); however, in the closed state, as long as the targeted bacteria are trapped in the NW arrays, these NWs are instantly triggered, bending to close the trap. Therefore, these bendable polycrystalline NWs simultaneously satisfy two requirements of an ideal dialyzer substrate: firstly, negligible deformation at high flow velocities to provide a high sensitivity to targeted microorganisms and secondly, significant bending at the tip region under weak carbohydrate–protein-binding forces to avoid desorption of immobilized bacteria by the shear stress of fluid bloodstream. Compared to unbendable single-crystalline NWs, such bendable polycrystalline NWs efficiently improve the number of captured bacteria in patient's bloodstream at various velocities, showing minimal interactions with other blood components.

## Results

**Fabrication and characterization of dialyzer.** Ordinarily, blood is considered a pathogenic bacteria-free zone. Once pathogenic bacteria enter the bloodstream, they begin producing antigens that are recognized by the immune system, triggering systemic inflammation, which can lead to multiorgan system failure, septic shock, and death (Fig. 1a)[22]. A hemodialysis device filled with NWs pre-grafted on CF is designed to remove microbial pathogens from patient's bloodstream (Fig. 1b and Supplementary Fig. 1). The backbone of the filter was prepared by directly carbonizing a melamine foam. Considering the size of an erythrocyte

(5–7 μm in diameter), a leukocyte (7–15 μm in diameter), and bacteria (*Salmonella*, rod of $0.5 \times 1.5$ μm), the ~200 μm pore size of CF was selected, which is approximately two orders of magnitude larger than the bacterial cells and one order than the blood cells (Fig. 1c and Supplementary Fig. 2). The original bloodstream rate was mainly maintained because of big pore aspect ratio of CF[23]. Subsequently, metal precursors (Ni, Co) were hydrothermally deposited on the CF to grow a high density of needle-like $NiCo(OH)_2CO_3$ NWs with a length of $5 \pm 0.5$ μm, a root diameter of $160 \pm 10$ nm, and a head diameter of $20 \pm 15$ nm (Fig. 1d, e and Supplementary Fig. 3). Their single-crystalline nature was confirmed by high-resolution transmission electron microscopy (HRTEM; Fig. 1f). The corresponding selected-area electron diffraction (SAED) pattern (Fig. 1g) shows sharp and clear Bragg spots. These pristine single-crystalline $NiCo(OH)_2CO_3$ NWs are converted into polycrystalline $NiCo_2O_4$ NWs by annealing at 300 °C for 2 h without noticeable alterations in morphology (Fig. 1h). About 28 wt% of $CO_2$ and $H_2O$ are released according to thermogravimetric analysis (TGA; Supplementary Fig. 4a). The 15–30 nm small crystal domains in the polycrystalline NWs (Fig. 1i, j) are consistent with the X-ray diffraction (XRD) analysis that reveals an average crystallite size of $20 \pm 4.5$ nm (Supplementary Fig. 4b). The corresponding SEAD pattern indicates typical polycrystalline diffraction rings, corresponding to the (220), (222), (400), and (440) planes (Fig. 1k).

**Nanomechanical properties of NWs.** Deformation in materials occurs most commonly due to the slip process of grain boundaries, which involves dislocation motions[24, 25]. Single-crystalline NWs have only one grain or crystal without grain boundaries. Consequently, the single-crystalline NWs clearly show higher moduli compared to the polycrystalline NWs. Representative atomic force microscopy (AFM) height images and their corresponding quantitative nanomechanical maps are shown in Fig. 2a–f, along with schematic representations of the nanoscale structures. The Young's modulus ($E$) values of single-crystalline and polycrystalline NWs were estimated according to the Derjaguin–Mueller–Toporov (DMT) model, in which their corresponding modulus distributions were collected along the contours of a single NW. Both showed decreases in $E$ with decreasing NW diameter, ranging from ~3 to ~15 GPa along the 30 nm head to the 160 nm root for an individual single-crystalline NW (Fig. 2g), and ~0.8 to ~1.3 GPa for a single polycrystalline NW (Fig. 2h). This phenomenon can be explained by the fact that the atomic coordination and cohesion near the surface in small-diameter NWs are relatively weaker vs. larger diameter ones[26]. Based on the hyperelastic model of the St. Venant–Kirchhoff theory (see Supplementary Methods), the relationship between the transverse force ($F$) and $x$-displacement ($\delta$) was derived (Fig. 2i), in which the NW shape was modeled as a truncated cone according to the TEM and scanning electron microscopy (SEM) results. When we simulated the bending performance of a single NW by a finite element method (FEM) analysis using the COMSOL Multiphysics software (ver. 5.2; COMSOL AB, Stockholm, Sweden), the relatively soft polycrystalline NWs preferred to be bent when a transverse force was applied. The deformation of polycrystalline NWs was significant under a pN transverse force.

Furthermore, the relationship between $F$ and $\delta$ was demonstrated by in situ bending experiments in an environmental SEM (ESEM) system equipped with a micromanipulator. An individual NW was pushed horizontally using a tungsten probe driven by a holder with a movable piezoelectric head. As shown in Fig. 2j, k, the polycrystalline NW had a larger bending curvature than that of the single-crystalline NW, in good agreement with the simulation (Fig. 2i). After releasing from pushing, the single-

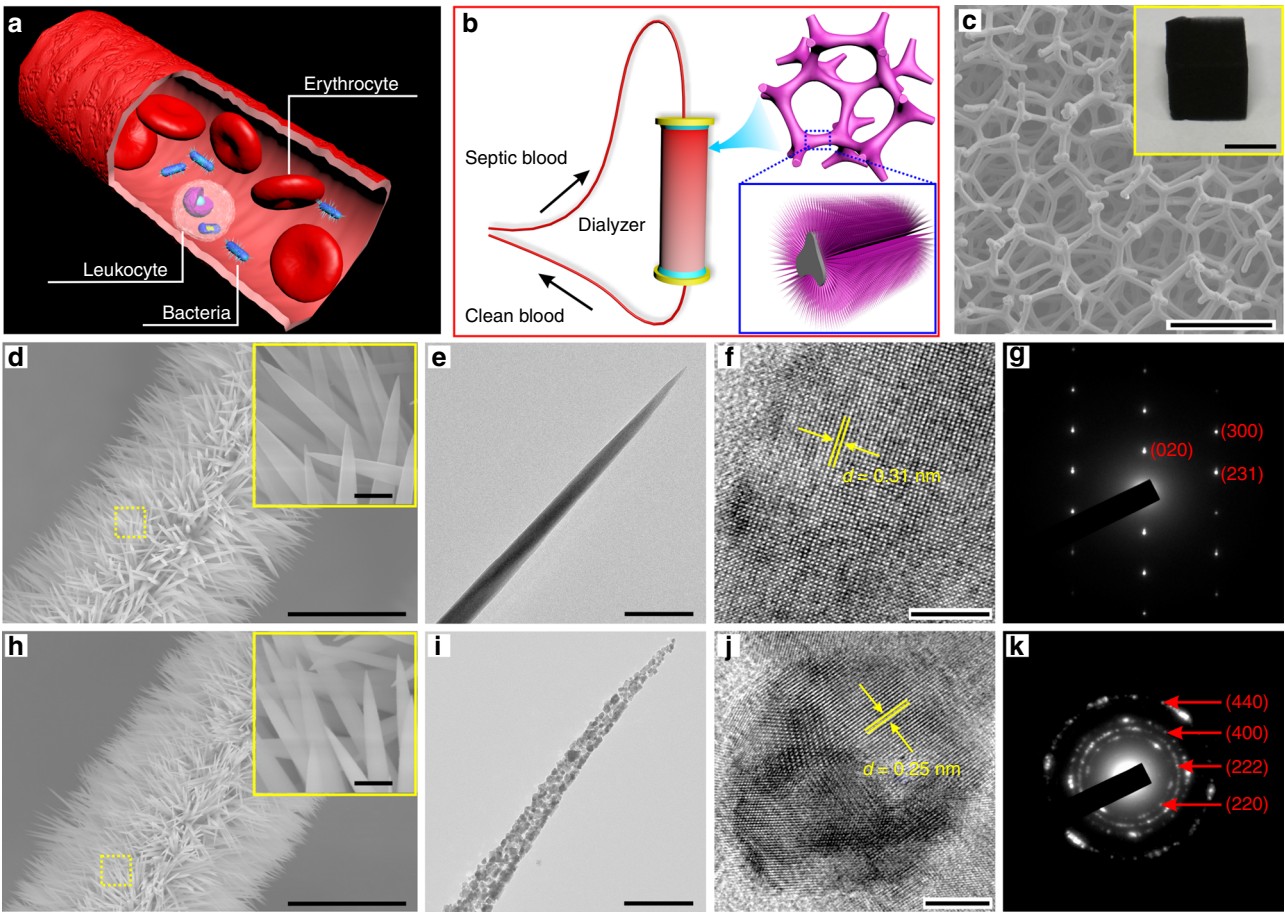

**Fig. 1** Fabrication and characterization of a dialyzer. **a** Schematic illustration of the complex blood environment in a blood vessel demonstrating the challenge of bacterial capture. **b** Blood cleansing by flowing the bacteria-contaminated blood through a dialyzer. **c** SEM image showing the internal structure of a dialyzer composed of NWs pre-grown on CF, and a photo of whole NWs/CF shown in the inset. Scale bars in **c** and inset are 250 μm and 1 cm, respectively. **d–g** Structural characterization of single-crystalline NWs. **d** Low- and high-magnification (inset) SEM images. **e** TEM image showing the NW has only one crystal domain. **f** HRTEM image and **g** responding SAED pattern demonstrate single crystal feature. **h–k** Structural characterization of polycrystalline NWs. **h** Low- and high-magnification (inset) SEM images. **i** TEM image of an individual $NiCo_2O_4$ NW showing multi-grain boundaries. **j** HRTEM image and **k** responding SAED pattern confirm the multi-crystal zones. Scale bars in **d** and **h** are 10 μm. Scale bars in the insets of **d** and **h** are 500 nm. Scale bars in **e** and **i** are 200 nm. Scale bars in **f** and **j** are 5 nm

crystalline NW displayed an elastic behavior by recovering its initial straight shape, while the polycrystalline NW showed plastic deformation, still maintaining its bent status due to its lower $E$ value.

**Formation conditions of 3D nanoclaws at high flow velocity.** Two aspects must be considered in the design of an ideal dialyzer with efficient capture: sorption and desorption of the bacteria. For clinical hemodialysis, soft NWs ($E < 1$ GPa), such as polystyrene nanotubes[14], polypyrrole NWs (0.53 MPa)[15], and PDMS microposts (2.5 MPa)[16], are seriously deformed by the high shear stresses of the fluid at 10 cm s$^{-1}$ velocity (Fig. 3a and Supplementary Fig. 5a). The bacteria are only adsorbed on the surfaces of bending nanostructures[16], and thus the bacteria will fall off readily because of losing landing site, resulting in a low bacteria capturing efficiency. Contrastively, although stiff NWs ($E > 2$ GPa), such as ZnO (~8 GPa)[27], gold (~70 GPa)[28], Si (~150 GPa)[29], and gallium phosphide (~150 GPa)[30], maintain their straight shape at high velocities, these adsorbed bacteria can dynamically be drove away from the surface/material by the shear force of the fluid, resulting in desorption of the bacteria (Fig. 3b). When the Young's modulus ranges from 1 to 2 GPa, such moderately stiff NWs cannot be bent by the shear force of the 10 cm s$^{-1}$ velocity fluid, but are deformed

by the ~100 pN molecular force of ligand-receptor (Fig. 3c). The strain-producing NW deformation as a function of stiffness or flow velocity was simulated by a computational fluid dynamics method, using a fluid structure interaction method and the incompressible Navier–Stokes equation (Supplementary Fig. 5b). Bloodstream moving along a CF pore will incur a shear stress, in which the fluid speed at the bottom of the NW is nearly zero and eventually increases to the speed at the tip region. The shear stress is proportional to the relative velocity. Therefore, significant deformation occurs at the tip region of NWs (Fig. 3d). For example, in the case of a flow rate of 10 cm s$^{-1}$ and an $E$ value of 0.06 GPa, von Mises stresses generated 1.65 μm displacement for a NW with a size consistent with the as-prepared NWs (Fig. 3e). The displacements of the prepared single-crystalline NWs and polycrystalline NWs correlated positively with the flow velocity (Fig. 3f). Because the shear stress arises from the force vector component perpendicular to the NW cross-section on which it acts, the NW deformation has an angle-dependent behavior with the fluid direction (Supplementary Fig. 5c–f).

**Characterization of 3D nanoclaws formation.** The present driving force for bacterial capture is mainly focused on molecular recognition such as carbohydrate–protein interaction and

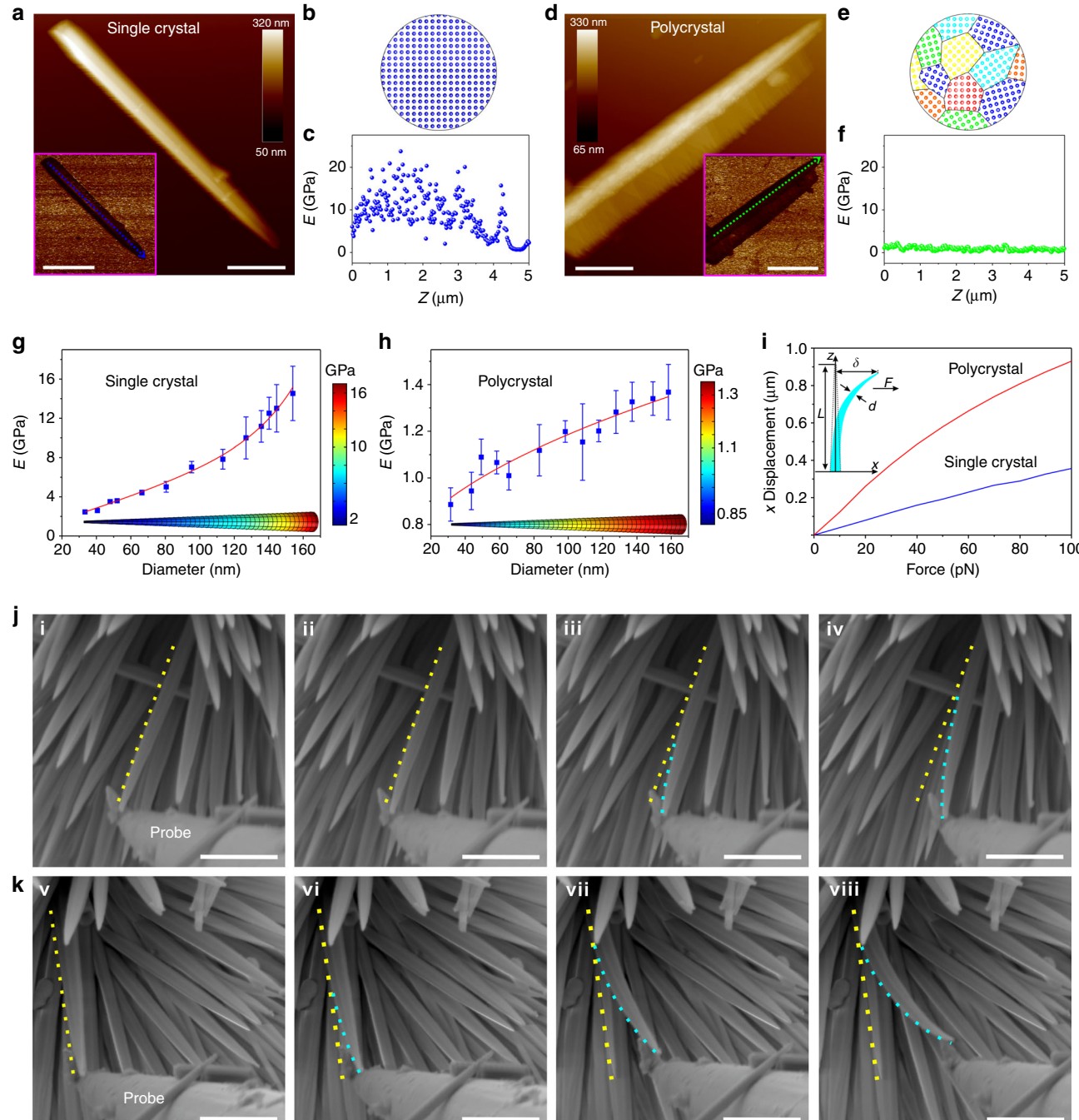

**Fig. 2** Nanomechanical properties of NWs. **a** Atomic force microscopy (AFM) height channel visualizing a single-crystalline NW. The inset shows the corresponding Derjaguin—Mueller—Toporov (DMT) Young's modulus map. Scale bars in **a** and inset are 1 and 2 μm, respectively. **b** Schematic representation of the NiCo(OH)$_2$CO$_3$ NW with one crystal domain. **c** Profile analysis of Young's modulus map along the blue dashed line. **d** AFM height channel visualizing a polycrystalline NW. The inset shows the corresponding DMT Young's modulus map. Scale bars in **d** and inset are 1 and 2 μm, respectively. **e** Schematic representation of the NiCo$_2$O$_4$ NW with nanoscale multi-crystal domain. **f** Profile analysis of the Young's modulus map along the green dashed line. **g**, **h** Experimental Young's modulus of NWs as a function of diameter size and simulated Young's modulus distributions along a single NW by COMSOL software. **g** Single-crystalline and **h** polycrystalline NWs. Error bars: standard error ($n = 3$). **i** Relationship between the applied force parallel and the lateral deflection distance ($x$ displacement) determined by FEM using the non-linear mode in the COMSOL software. The inset shows the geometric parameters for a bent NW depicting the deflection ($\delta$) when a force ($F$) is applied at the top. The bending properties of NWs in situ monitored by environmental SEM, **j** single-crystalline (i to iv) and **k** polycrystalline NWs (v to viii). Scale bars in **j** and **k** are 500 nm

antigen–antibody binding. Concanavalin A (Con A) and *Salmonella*, pathogenic Gram-negative bacteria, which cause various illnesses including gastroenteritis and systemic febrile disease[31], were selected for a molecular recognition model. The driving

force was generated by specific lectin–carbohydrate recognition between Con A and mannose of the bacterial surface lipopolysaccharide (Fig. 4a, b and Supplementary Fig. 6), which is ~100 pN[32–34]. This strong adhesion force enables the deformation of

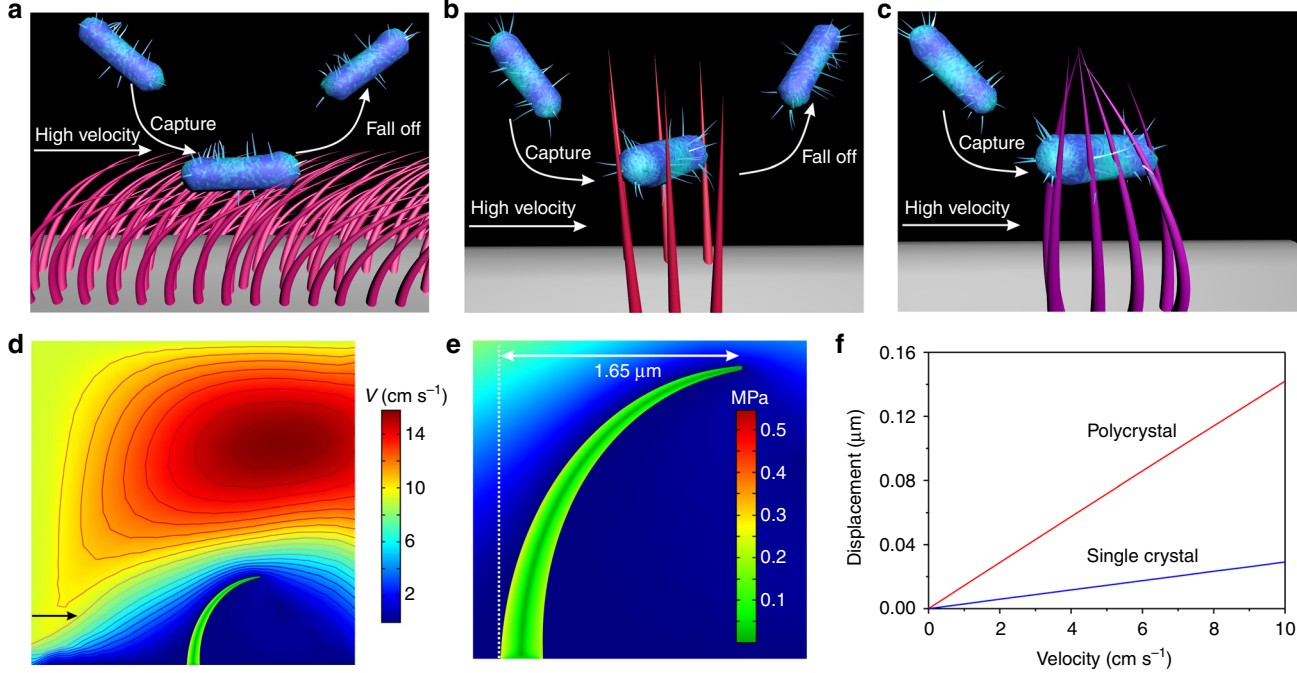

**Fig. 3** Analysis of 3D nanoclaws formation conditions. **a–c** Three models of bacterial capture at high flow velocity depending on the $E$ of the NWs. **a** The bacteria loaded on bending surface of soft NWs ($E < 1\,\text{GPa}$) will fall off readily. **b** Adsorbed bacteria can dynamically desorb from stiff NWs ($E > 2\,\text{GPa}$) surface as the shear force of the blood stream. **c** 3D nanoclaws generated by bendable moderate NWs ($1\,\text{GPa} < E < 2\,\text{GPa}$) prevent the immobilized bacteria from washing away at high flow velocity. **d** The flow field around a single NW, wherein the arrow indicates the flow direction and the contour lines represent fluid flow velocity of different levels. **e** The magnified region of flow distribution illustrates that the von Mises stresses concentrate at the bottom of a single NW. The deformation of a NW ($E = 0.06\,\text{GPa}$) is $1.65\,\mu\text{m}$ at $10\,\text{cm s}^{-1}$ velocity. **f** Comparison of the deformation of single-crystalline and polycrystalline NWs under different flow velocities

polycrystalline NWs when the bacteria are trapped in NW arrays. To demonstrate this, parallel experiments were performed to replace the strong molecular recognition by a relatively weak electrostatic interaction force, in which polyethylenimine (PEI) was modified on the NW surface to capture negatively charged bacteria[35] (Supplementary Fig. 7 and Fig. 4c, d). The successful modification of NWs with Con A was evidenced by fluorescence detection using rhodamine B-labeled Con A (Supplementary Fig. 8). The bacterial capture performance was evaluated by a homemade dialyzer filled with CF, and single-crystalline NWs/CF and polycrystalline NWs/CF (Supplementary Fig. 9). Compared to PEI, conjugating Con A significantly improved the bacteria capturing ability, from $15.1 \pm 1.9 \times 10^8\,\text{CFU cm}^{-3}$ on bare single-crystalline NWs/CF, and $19.1 \pm 2.2 \times 10^8\,\text{CFU cm}^{-3}$ on PEI-single-crystalline NWs/CF, to $42.2 \pm 2.2 \times 10^8\,\text{CFU cm}^{-3}$ on Con A-single-crystalline NWs/CF. Similarly, this enhancement effect was also observed in polycrystalline NWs/CF, from $17.9 \pm 3.1 \times 10^8\,\text{CFU cm}^{-3}$ on bare, and $27.4 \pm 2.6 \times 10^8\,\text{CFU cm}^{-3}$ on PEI coated, to $84.2 \pm 1.8 \times 10^8\,\text{CFU cm}^{-3}$ on Con A conjugated (Fig. 4e). The bacterial capture amount is the largest on Con A-polycrystalline NWs/CF, which is attributed to the formation of nanoclaws by average four bending polycrystalline NWs (Supplementary Fig. 10).

The polycrystalline NW deformation induced by the strong adhesion force of Con A was confirmed by SEM, in which FEM analysis was performed to gain an insight into the NW deformation. The single-crystalline NWs nearly maintained their pristine status (Fig. 4f, g), whereas the polycrystalline NWs showed a ~1 μm bend (Fig. 4h, i). The lateral deflection distance of NWs changed with diameter size (Supplementary Fig. 11), and the NW with a sharp tip showed a more metamorphous nature

than that of a blunt NW under a constant and weak driving force of 100 pN. The von Mises stresses were concentrated at the NW tip, resulting in 100 nm displacement for single-crystalline NWs and 1 μm deformation at the tips of polycrystalline NWs (Supplementary Fig. 12). The driving force increased almost linearly as $E$, ranging from 0.5 to 50 GPa (Supplementary Fig. 13), which suggests that the polycrystalline NWs could create more significant deformation when increasing the driving force.

A 1-μm deflection distance of the polycrystalline NWs was sufficient to prevent the trapped bacteria from desorbing, which was examined by the washing processes at high flow rates using 4′-6-diamidino-2 phenylindole (DAPI) stain. Before washing, the polycrystalline NWs/CF dialyzer had the highest bacterial capture capacity (Fig. 5a). After washing with sterile normal saline at 20 cm s$^{-1}$ for 2 min, 85% of the bacteria in the polycrystalline NWs/CF dialyzer were still held, while only 35% and 5% of residual bacteria were retained on a single-crystalline NWs/CF dialyzer and a smooth CF dialyzer (Fig. 5b, c and Supplementary Fig. 14). The shear force effect on bacterial capturing capabilities was assessed by a peristaltic pump that controlled the flow velocities (Fig. 5d). The numbers of captured bacteria in the dialyzers filled with single-crystalline or polycrystalline NWs/CF reached a maximum at 10 cm s$^{-1}$. At low flow velocities (2.5, 5, 7.5 cm s$^{-1}$), the behavior of the two dialyzers was comparable, but bendable NWs represented significant advantage at higher flow velocities (10, 12.5, 15 cm s$^{-1}$). Increasing the bacterial capture ability of two dialyzers from 2.5 to 10 cm s$^{-1}$ of flow velocity could be attributed to two aspects: the first is to bring more bacteria to the surface because of increasing total volume to flow through, and the second is to increase the collision frequency between bacteria and NWs.

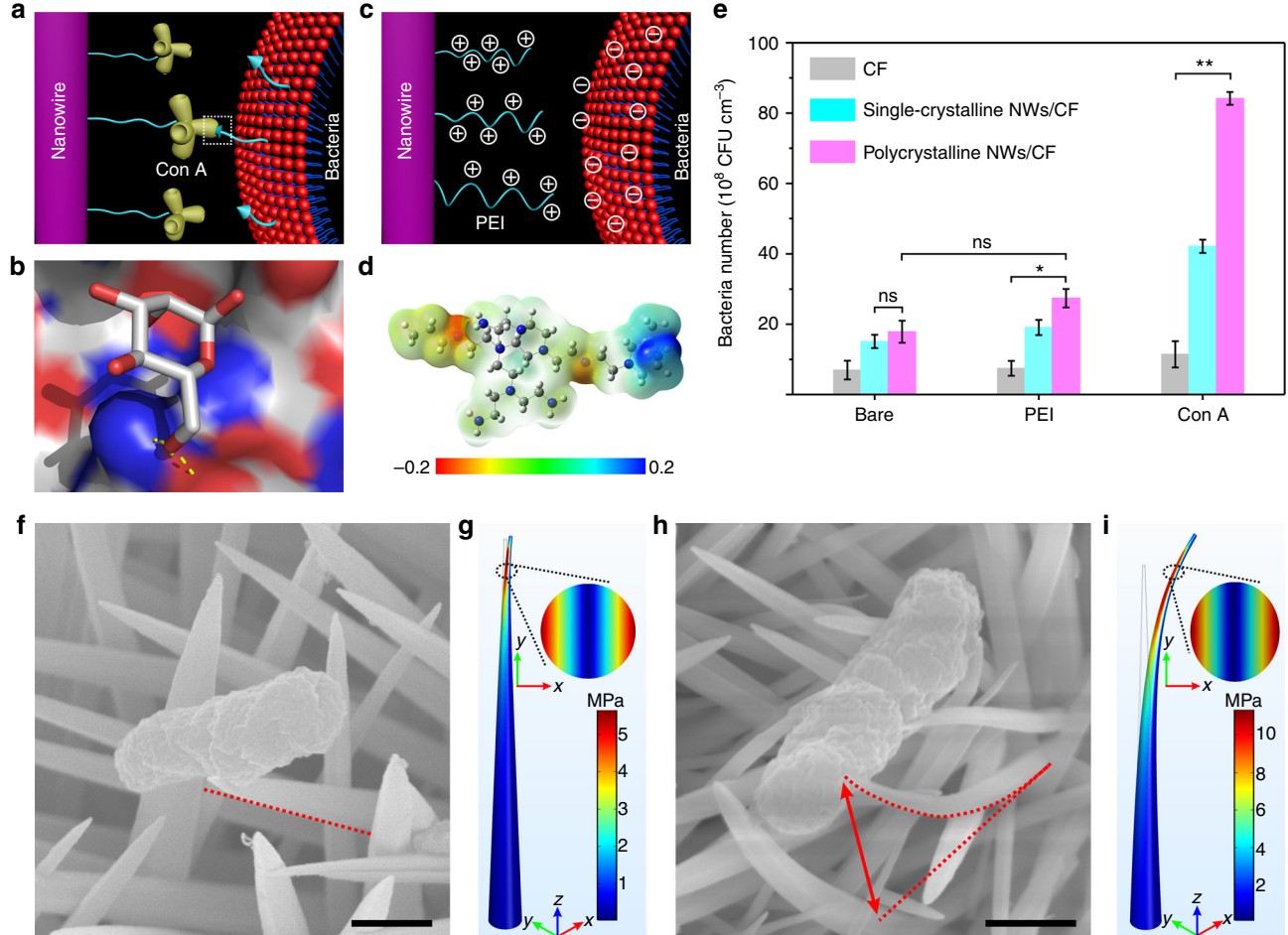

**Fig. 4** Characterization of 3D nanoclaws formation. **a** Con A on NW surfaces bound to mannose on the bacterial membrane. **b** The graph shows the hydrogen bonds (yellow sticks) in the crystal structure of the binding site of mannose to Con A (PDB code 1i3h) using the PYMOL software. **c** Illustration of electrostatic interactions between the cationic polymer of PEI modified on NW surfaces and bacterial membranes. **d** An electrostatic potential map for a representative conformation of PEI. **e** Comparing the bacterial capturing capacity of different molecules modified on NW surfaces at a flow velocity of 10 cm s$^{-1}$. Error bars: standard error ($n = 3$). Student's $t$-test, **$P < 0.01$; ns, not significant. **f–i** SEM measurements and FEM simulations for the deflection distance of NWs after bacterial capture, **f** single-crystalline and **h** polycrystalline NWs of SEM images. The von Mises stresses of **g** a single-crystalline and **i** a polycrystalline NW concentrated at the NW tips under constant displacements of 100 nm and 1 μm, respectively. A 2D diagram corresponding to the von Mises stress at a height of 4.5 μm (inset). Scale bars in **f** and **h** are 500 nm

**Bacterial capture performance in bloodstream**. To assess the capture capabilities in a practical system, adult human blood containing $1.0 \times 10^8$ CFU mL$^{-1}$ bacteria was passed through the dialyzers filled by CF, single-crystalline NWs/CF, and polycrystalline NWs/CF at 10 cm s$^{-1}$. The bacterial capture efficiency on CF, single-crystalline NWs/CF, and polycrystalline NWs/CF were ~10, ~40, and ~97%, respectively (Fig. 5e), which was similar to the sterile normal saline (Supplementary Fig. 15a), indicating a favorable performance in the complex bloodstream environment. Moreover, there is no marked difference in bacterial capture ability whether the anticoagulants were added or not (Supplementary Fig. 15b). The residual bacteria were further confirmed by growing on agar plates (Fig. 5f–h). Non-specific adsorption of human blood components (red blood cells, white blood cells, and platelets) were rarely observed on the NWs/CF dialyzers (Supplementary Fig. 16). As shown in Supplementary Fig. 17, the damage of sharp NWs for blood cells is negligible, which could be attributed to two reasons. First, the pore size of CF (~200 μm) is large enough to allow the blood cells smoothly through the dialyzer. Second, the NWs are coated with Con A that binds specifically to the surfaces of clinical pathogens, rather than blood cells. Additionally, an important

consideration for clinical implementation of a dialyzer is minimization of the time that the blood spends outside the body, to avoid coagulation or infection. The flow rate in our system can reach as high as 50 mL min$^{-1}$ (Eq. 2 in the Methods), which is much higher than that in previously reported methods, such as the biospleen device (0.167 mL min$^{-1}$)[36], a track-etched polycarbonate filter (0.667 mL min$^{-1}$)[37], and magnetic nanoparticles (1.0 mL min$^{-1}$)[9].

## Discussion

Our data show that bendable polycrystalline NWs can improve the bacterial capturing efficiency of the dialyzer to patient's bloodstream. One would expect nanomaterials to offer an excellent opportunity to improve the capture sensitivity of biomolecules due to their high surface area-to-volume ratio and similar size, but ignoring the immobilized biomolecule desorption due to the high shear forces of the fluid bloodstream. The observed increase in the bacterial capture efficiency in the bloodstream at a 10 cm s$^{-1}$ velocity, from ~10% on CF and ~40% on unbendable single-crystalline NW/CF to 97% on bendable polycrystalline NW/CF, is significant because the bent NWs should increase the

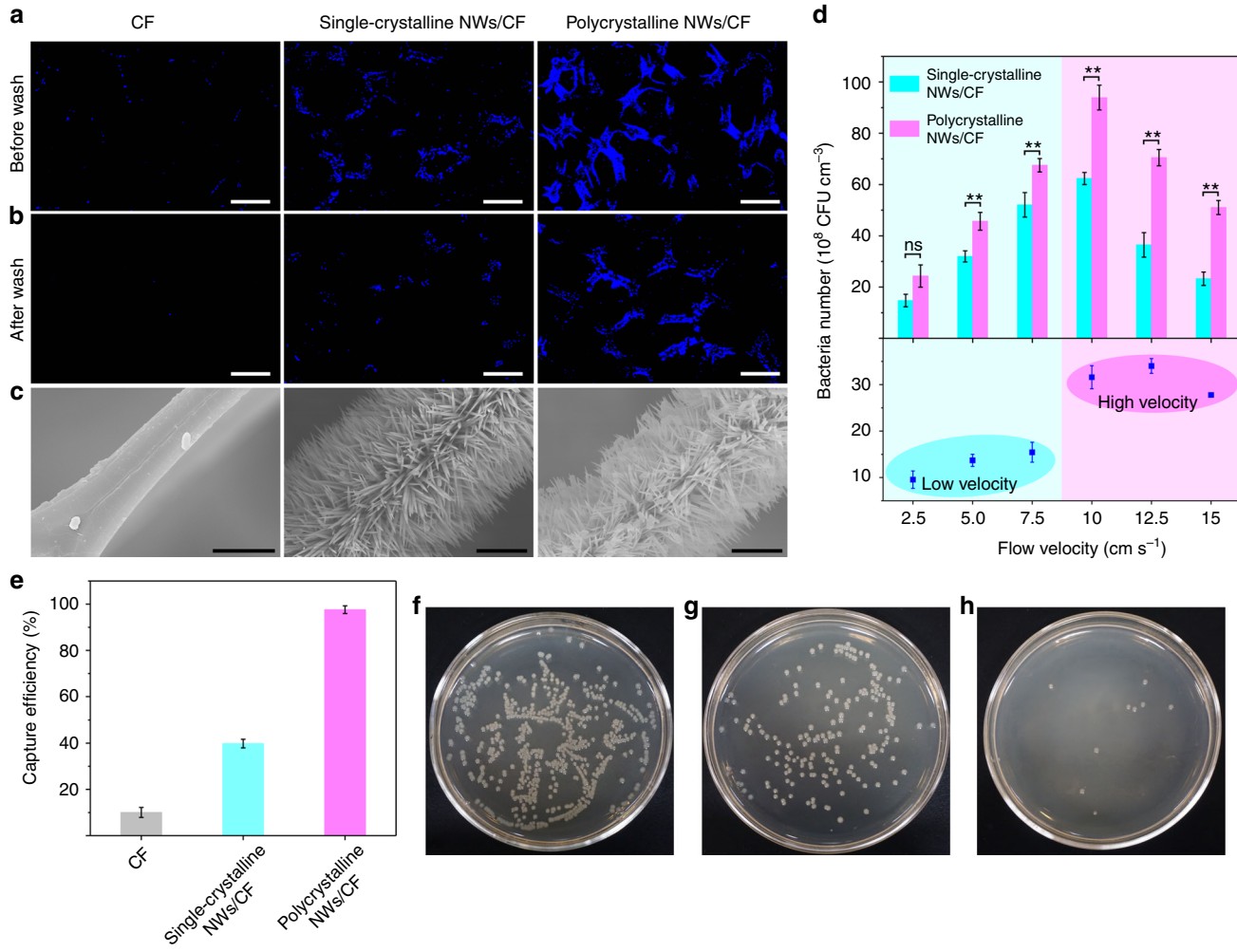

**Fig. 5** Bacteria capturing performance in the bloodstream. Fluorescent images of bacteria captured in three dialyzers **a** before and **b** after washing with sterile normal saline at a flow velocity of 20 cm s$^{-1}$ for 2 min. The bacteria were stained with DAPI shown in blue. Scale bars in **a** and **b** are 100 μm. **c** Corresponding SEM images of the dialyzers after washing. Scale bars, 5 μm. **d** Quantitative evaluation of the number of captured bacteria using Con A-modified dialyzers at different flow velocities, and the difference in the number of bacteria captured between the dialyzers. **e** Bacterial capturing efficiencies of the three dialyzers. Bacteria spiked into adult human blood and flowed through the dialyzers at 10 cm s$^{-1}$. **f–h** Photograph of an agar plate visualizing the residual bacteria among three dialyzers, including **f** CF, **g** single-crystalline NWs/CF, and **h** polycrystalline NWs/CF. Error bars: standard error ($n = 3$). Student's $t$-test, **$P < 0.01$; ns, not significant

anti-shearing force of the captured bacteria. Such deformation of polycrystalline NWs was understood in terms of the simulated and measured Young's modulus distribution along a single NW. The demonstration of efficient bacterial capture in normal saline and the human bloodstream, in a dialyzer filled by bendable polycrystalline NWs/CF, is clearly a major step toward the development of a nanotechnology platform that can meet evolving clinical and lifestyle needs. In principle, other microorganisms, such as viruses, circulating cancer cells, and stem cells, could be used according to this proof-of-concept to create a nanobiotechnology platform for the fabrication of multifunctional artificial kidneys.

## Methods

**Preparation of CF**. The CF was prepared by carbonizing melamine foam (MF) under an argon atmosphere. MF was first cut into $1.2 \times 1.2 \times 0.8$ cm$^3$ pieces before carbonization. Carbonization used a previously reported method[38, 39]. First, the temperature was raised from room temperature to 300 °C, at a rate of 5 °C min$^{-1}$, and then held constant for 5 min. Second, the temperature was further raised to 400 °C at a rate of 1 °C min$^{-1}$ and held for 5 min. Finally, the temperature was

raised to 700 °C at a rate of 2 °C min$^{-1}$ and held for 120 min. The as-prepared CF was removed once the temperature was below 80 °C.

**Synthesis of NiCo(OH)$_2$CO$_3$ NWs/CF and NiCo$_2$O$_4$ NWs/CF**. In a typical process, 2 mmol of CoCl$_2$·6H$_2$O, 1.25 mmol of NiCl$_2$·6H$_2$O, and 3 mmol of urea were dissolved in 15 mL of water to form a transparent pink solution. A piece of carbonized CF ($0.8 \times 0.8 \times 0.3$ cm$^3$) was placed in the solution, and the solution was then transferred to a 25 mL Teflon-lined stainless steel autoclave and held at 120 °C for 6 h. After hydrothermal growth, the NiCo(OH)$_2$CO$_3$ NWs/CF was washed carefully with deionized water and ethanol several times to remove the excess surfactant and dissociative ions, and then finally dried in air. To obtain the NiCo$_2$O$_4$ NWs/CF, the NiCo(OH)$_2$CO$_3$ NWs/CF sample was placed in a quartz tube and annealed at 300 °C for 2 h to obtain well-defined crystallized NiCo$_2$O$_4$ NWs on CF.

**Bacterial filtrating and staining**. For bacterial filtering, Con A-functionalized pieces having dimensions $0.4 \times 0.4 \times 0.3$ cm$^3$ of CF, NiCo(OH)$_2$CO$_3$/CF, and NiCo$_2$O$_4$/CF were fixed on a homemade filtration device. Then, the bacterial stock solutions were passed through the substrates at a flow rate of 50 mL min$^{-1}$. After filtering, the substrates were removed from the device and dried in an argon atmosphere prior to fluorescent staining. The bacterial capture number ($N_c$) of the substrates was defined as the number of bacteria captured before and after filtering

and calculated as follows:

$$N_c = \frac{V \times (C_a - C_b)}{V_s},\tag{1}$$

where $C_a$ and $C_b$ represent the concentration of bacteria in the test solution before and after filtering, respectively, and $V$ and $V_s$ are the volumes of the test solution and substrate, respectively. The bacterial concentrations were determined using the standard colony counting method in our experiments.

For bacterial staining, the substrate with captured bacteria was first incubated with DAPI (5 µg mL$^{-1}$) for 15 min in the dark, and then rinsed three times with sterile normal saline before microscopy imaging.

The flow rate, $Q$, was calculated using the following equation:

$$Q = v \times \pi r^2,\tag{2}$$

where $v$ and $r$ are the flow velocity and radius of the tube (0.16 cm), respectively.

**Bacterial capture from bloodstream.** The bacteria ($1.0 \times 10^8$ CFU mL$^{-1}$) were first spiked into healthy adult human blood ($5.0 \times 10^7$ mL$^{-1}$), which was pretreated with anticoagulants (heparin) to prepare artificial septic blood samples. After dialyzer separation, the number of bacteria were quantified using the standard plate count method in an appropriate culturing agar medium. Blood cells were counted using a conventional blood counting chamber. Blood cell viability was determined by using calcein acetoxymethyl ester (AM) assay[40]. Calcein-AM was added to the blood at the final concentration of 1 mM and the cells were incubated at 37 °C in the dark for 40 min. The fluorescence was read at an excitation wavelength of 490 nm and an emission wavelength of 520 nm using a microwell plate reader. The relative cell viability was obtained by comparing the fluorescence signals of pre-and post-filtration.

**Data availability**. The authors declare that all data supporting the findings of this study are available within the article and its Supplementary Information files or from the corresponding author on reasonable request.

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

## Acknowledgements

T.W. acknowledges the support from the 1000 Young Talents program, the National Natural Science Foundation of China (Grant Nos. 21321003, 21635002, 21621062), and ICCAS. J.L.W. acknowledges the support from the New Century Excellent Talents in University (NCET-13-0483) and National Natural Science Foundation of China (No. 31371813, No. 31201357, and No. 21675127).

## Author contributions

L.Z.L. and T.W. conceived and designed the project. L.Z.L. carried out the synthesis and performed all the experiments. S.C. performed the FEM simulations. Z.J.X. assisted the AFM test. Z.Z. conducted HRTEM characterization. X.Z.Q. assisted the TEM characterization. Z.X.N. offered the bacteria culture platforms. D.H. assisted the in situ ESEM

characterization. L.Z.L., J.L.W., and T.W. co-wrote the manuscript. All authors discussed the results and commented on the manuscript.

## Additional information

**Competing interests:** The authors declare no competing financial interests.

