## [Peer Review File · Nature Communications]

Reviewers' comments:

Reviewer #1 (Remarks to the Author):

The study by Liu et al developed a very smart way of capturing bacteria using polycrystalline nanowires, which they name nanoclaws. They took advantage of the moderate level of Young's modulus at the tip of polycrystalline nanowires as opposed to their single-crystalline counterparts, a property that was characterized systematically through simulations and experiments. I commend that the authors took systematic steps to go from method development to physical characterization to medical applications, and indeed prove that the approach is advantageous in comparison to a number of current methods. I find it an elegant piece of work and recommend it for publication. I do find, however, a number of minor issues, despite not affecting the general story line of the paper, are best to be addressed before publication.

1. I suggest that the authors shorten the intro part in the abstract and describe more clearly what approach they have developed. I recommend that the authors separate the long sentence 'Here.....' into two sentences, which will facilitate the readers to easily catch 1) what it does, and 2) how it works. Current one is a bit too complicated.

2. As authors also noted that the blood cells and bacteria are in different sizes, perhaps the authors can be more specific on the numbers. Bacteria can also come in different sizes and shapes. White blood cells are 12-15 micron in diameter, which is 1 order of magnitude, but not 2 orders of magnitude smaller than the 200 micron pore size in the CF. An E. coli can be 1x3 microns a rod, and how big are the salmonella they used? How does shape or surface-to-volume ratio change capture efficiency? Bacteria can perhaps be smaller in Fig. 1a.

3. it is not entirely apparent how simply by addressing a large pore size would lead to a maintenance of bloodstream rate. I suppose It might depend on pore grid arrangement, viscosity, and cell density etc. Can the author provide a reference with numerical simulations, or better provide experimental data to show no slowing down of blood flow with the CF was observed?

4. Authors sometimes write 'introduction'-like text in the results section, such as line 98-101, which can be helpful for the readers, but please also reference where the statements come from. For example, here, the difference in bending stiffness between single- and poly-crystalline.

5. 'Molecular force' is a very generic term, at least I find it not well defined, can the authors be more specific? They have used conA-maltose as a combination for capture, it would be great to extend the discussion on how broadly application such method is, and what can be the alternative approaches that would fit the '100pN' force range for similar or different method. Are forces exerted by chemical bonds restrained to ~100pN for the method to work? or can it be bigger? perhaps can discuss in the text.

6. Line 73, Lipopolysaccharides are naturally on the bacterial surface. They don't necessarily get 'secreted' in order to become antigen when entering the blood stream.

7. Change all 'absorb' and 'absorption' to 'adsorb' and 'adsorption'.

8. Mind very long sentences, like line 177-183. If they are more than 3 lines, often it is a good sign of breaking them up.

9. Line 137-144, the writing should be improved for clarity.

10. Line 196-198. Watch the past tense.

11. Fig. 5a/b, blue is rather hard to visualize. best to show in greyscale to reflect the difference.

12. Fig. 5d, bottom panel, I think it is better to show difference in percentile.

13. What is the effect of high flow rate? Why do the devices trap more cells at high flow rate? Is it simply because of the total volume that was flown through or because of the profile of the flow that brings bacteria to the surface?

14. Reference Fig. 13 for the 'survival' of the blood cells in the last result section. It is interesting that blood cells were able to survive the sharp ends of the nanowires, given that they do not have cell wall. I also suppose that with different cell sizes, blood cells and bacteria can have different probability of being close to the grid that contains nanowire, can the authors comment on that briefly?

15. Line 61: MW -> NW

16. Can authors quantify or have an estimate over what the contact surface for the capture, from their SEM images? (or easier, count how many nanowire per bacteria).

-Fabai Wu

Reviewer #2 (Remarks to the Author):

The Nanowire technology is interesting but you have not proved to me that you can actually capture bacteria from blood using your system without damage to the blood cells.

I have the following comments:

1. you have not reviewed the literature sufficiently - there are many groups who have successfully removed pathogens at higher flow rates (>100 mL/min) than you quote e.g. McCrea K et al (2014) Removal of Carbapenem-Resistant Enterobacteriaceae (CRE) from Blood by Heparin- Functional Hemoperfusion Media. PLoS ONE 9(12): e114242. Didar et al (2015) Improved treatment of systemic blood infections using antibiotics with extracorporeal opsonin hemoadsorption Biomaterials. 2015 Oct;67:382-92.
2. You have focused the paper on the nanowire technology, but in the blood experiments, you add ConA - which captures pathogens by lectin affinity, not by the 3D nanoclaws on which you base the paper.
3. You do not give sufficient detail on the preservatives used to prevent blood clotting, you have not tested for clotting or coagulation activation in running the septic blood through the filter, you have not made control filters.
4. your claims are too strong, based on the single pathogen you have spiked into blood - the blood itself will clear > 90% of the pathogen by phagocytosis (depending on the preservative used and the temperature)

Reviewer #3 (Remarks to the Author):

This paper describes new removal method of bacteria in bloodstream. Bendable nanowires modified with Con A indicates high bacterial capture efficiency. Especially reviewer interested in high capture efficiency by appropriate nanowire with Young's moduli. The results are clear and explain (discussion) is enough. This paper will be interested in the wide field of medical field or material science and micro/nano material science. And the concept will influence micro/nano material design.

Minor comments:

Comment 1:

P4, lines 137, "Such NW deformation has a negative effect on the bacteria-capturing capacity. The bacteria are only absorbed on the surfaces of bending nanostructures¹⁵, resulting in a low bacteria capturing efficiency." These sentences are important for topic (bendable nanowires) of this paper. However, reviewer could not understand very well (immediately). Please insert below words in text because easy to understand. Or please explain more detail in text.

The bacteria are only absorbed on the surfaces of bending nanostructures, "and thus the bacteria will fall off readily". (although already written in Figure 3a caption)

Comment 2:

SEM images of the surface single-crystalline NWs/CF, polycrystalline NWs/CF at Figure 5c or supplementary Figure 13d and 13e are too small. Reader will be interested in the images of the adsorbed bacteria or blood cells. Please add expansion images at supplementary Fig 13.

Comment 3:

Reader cannot immediately understand what blue fluorescence indicate.

Please add words ex. "(Blue: DAPI)" at Figure 5.

Comment 4:

Have you put the anticoagulants in blood? If use, please describe in material and method.

To Reviewer #1:

General comment: The study by Liu et al developed a very smart way of capturing bacteria using polycrystalline nanowires, which they name nanoclaws. They took advantage of the moderate level of Young's modulus at the tip of polycrystalline nanowires as opposed to their single-crystalline counterparts, a property that was characterized systematically through simulations and experiments. I commend that the authors took systematic steps to go from method development to physical characterization to medical applications, and indeed prove that the approach is advantageous in comparison to a number of current methods. I find it an elegant piece of work and recommend it for publication. I do find, however, a number of minor issues, despite not affecting the general story line of the paper, are best to be addressed before publication.

Reply: We very much appreciate the reviewer's highly encouraging comments and very positive recommendation of our work. The corrections were highlighted by yellow in the revised manuscript.

Comment 1: I suggest that the authors shorten the intro part in the abstract and describe more clearly what approach they have developed. I recommend that the authors separate the long sentence 'Here.....' into two sentences, which will facilitate the readers to easily catch 1) what it does, and 2) how it works. Current one is a bit too complicated.

Reply: Agree, the abstract text was shorten to remove introduction part, and the long sentence was separated into two sentences as your suggestion (line 18 of page 1).

Comment 2: As authors also noted that the blood cells and bacteria are in different sizes, perhaps the authors can be more specific on the numbers. Bacteria can also come in different sizes and shapes. White

blood cells are 12-15 micron in diameter, which is 1 order of magnitude, but not 2 orders of magnitude smaller than the 200 micron pore size in the CF. An E. coli can be 1x3 microns a rod, and how big are the salmonella they used? How does shape or surface-to-volume ratio change capture efficiency? Bacteria can perhaps be smaller in Fig. 1a.

Reply: Agree, modified statements were added on line 34 of page 2 in revised manuscript.

The size of salmonella is around 0.5×1.5 μm (Fig. R1), which is smaller than E. coli. Although the effect of the bacterial shape (rod or sphere) and size on bacterial capture efficiency had not been observed in our experiments, interestingly, the nanowire's density, modification and Young's modulus will be re-designed to capture multi-bacteria because of considering the interaction force between bacteria and nanowires.

In addition, we greatly appreciate the suggestion for decreasing the size of bacteria in Fig 1a. We have corrected the scheme based on your suggestions.

Fig. R1 for reviewers only. SEM image of salmonella shown the typical rod-like shape.

Comment 3: it is not entirely apparent how simply by addressing a large pore size would lead to a maintenance of bloodstream rate. I suppose It might depend on pore grid arrangement, viscosity, and cell density etc. Can the author provide a reference with numerical simulations, or better provide experimental data to show no slowing down of blood flow with the CF was observed?

Reply: Agree, we modified the statement on line 39 of page 2, and provided a new reference about numerical simulations as ref 24.

Comment 4: Authors sometimes write 'introduction'-like text in the results section, such as line 98-101, which can be helpful for the readers, but please also reference where the statements come from. For example, here, the difference in bending stiffness between single- and poly-crystalline.

Reply: Agree, the relative papers were added in and cited as ref. 25 and 26.

Comment 5: 'Molecular force' is a very generic term, at least I find it not well defined, can the authors be more specific? They have used conA-maltose as a combination for capture, it would be great to extend the discussion on how broadly application such method is, and what can be the alternative approaches that would fit the '100pN' force range for similar or different method. Are forces exerted by chemical bonds restrained to ~100pN for the method to work? or can it be bigger? perhaps can discuss in the text.

Reply: The molecular force used in this manuscript indicates the specific binding of ligand-receptor (Con A-mannose). The molecular force were replaced by molecular force of ligand-receptor.

The nanowires are bent by molecular force of ligand-receptor, which are strongly depended on the stiffness of NWs and the driving force. As shown in supplementary Fig. 13, the driving force to bend nanowires is almost linear as E , which means weak force easily curves the soft nanowires and large force enable to bend the rigid nanowires. Because of the molecular recognition forces, such as carbohydrate-protein interaction [Nano Lett. 12, 396-401 (2012)] and antigen-antibody binding [ACS Nano 9, 5051-5062 (2015)] were usually in pN range [Nano Lett. 16, 1299-1307 (2016)], we selected the polycrystalline nanowires with ~ 1 GPa stiffness in this manuscript. Increasing the driving forces could induce more significant deformation of the nanowires (line 25 of page 5), which ensured the method to work efficiently.

In principle, other microorganisms, such as viruses, circulating cancer cells, and stem cells, could be used according to this proof-of-concept to create a nano-biotechnology platform for the fabrication of multifunctional artificial kidneys (line 41 of page 6).

Comment 6: Line 73, Lipopolysaccharides are naturally on the bacterial surface. They don't necessarily get 'secreted' in order to become antigen when entering the blood stream.

Reply: Agree, the "lipopolysaccharides" was replaced by "toxins" on line 30 of page 2.

Comment 7: Change all 'absorb' and 'absorption' to 'adsorb' and 'adsorption'.

Reply: The errors were corrected and marked by yellow highlight.

Comment 8: Mind very long sentences, like line 177-183. If they are more than 3 lines, often it is a good sign of breaking them up.

Reply: We greatly appreciate the suggestion for breaking long sentences into two short sentences, which were highlighted by yellow on line 7 of page 5.

Comment 9: Line 137-144, the writing should be improved for clarity.

Reply: We re-wrote the sentences to improve the readability on line 9 of page 4 .

Comment 10: Line 196-198. Watch the past tense.

Reply: The "is" was changed to "was".

Comment 11: Fig. 5a/b, blue is rather hard to visualize. best to show in greyscale to reflect the difference.

Reply: The reason we selected blue color is near to real vision, because the bacteria have been stained by typical blue DAPI. In order to improve visualization as you mentioned, we adjusted the contrast and brightness of Fig 5a and 5b.

Comment 12: Fig. 5d, bottom panel, I think it is better to show difference in percentile.

Reply: The purpose of Fig 3d bottom panel is to represent the difference between low and high velocity zones. The numbers of captured bacteria look more significant to distinguish the difference of two zones than using the percentile.

Comment 13: What is the effect of high flow rate? Why do the devices trap more cells at high flow rate? Is it simply because of the total volume that was flown through or because of the profile of the flow that brings bacteria to the surface?

Reply: Agree, we think the effect of high velocity flow on captured bacteria could be attributed to two aspects: the first is to bring more bacteria to the surface because of increasing total volume to flow through, and the second is to increase the collision frequency between bacteria and nanowires (line 41 of page 5).

Comment 14: Reference Fig. 13 for the 'survival' of the blood cells in the last result section. It is interesting that blood cells were able to survive the sharp ends of the nanowires, given that they do not have cell wall. I also suppose that with different cell sizes, blood cells and bacteria can have different probability of being close to the grid that contains nanowire, can the authors comment on that briefly?

Reply: To verify that blood cells were able to survive the sharp ends of the nanowires, we used calcein-AM to determine the cell viability. The experimental details were add on line 10 of page 12.

As shown in Supplementary Fig. 17, the damage of sharp NWs for blood cells is negligible, which could be attributed to two reasons. First, the pore size of CF (~200 μm) is large enough

to allow the blood cells smoothly through the dialyzer. Second, the NWs are coated with Con A that binds specifically to the surfaces of clinical pathogens, rather than blood cells (line 14 of page 6).

Comment 15: Line 61: MW -> NW

Reply: The "MW" was changed to "NW" on line 18 of page 2.

Comment 16: Can authors quantify or have an estimate over what the contact surface for the capture, from their SEM images? (or easier, count how many nanowire per bacteria).

Reply: It is a great suggestion to count how many nanowires per bacteria. Based on SEM images, the number of nanowires to form "nanoclave" could be average 4 after counting nearly 100 captured bacteria (Supplementary Fig. 10), which were highlighted by yellow on line 14 of page 5.

To Reviewer #2:

General comment: The Nanowire technology is interesting but you have not proved to me that you can actually capture bacteria from blood using your system without damage to the blood cells.

Reply: We sincerely appreciate to this reviewer for his/her recommendations and constructive suggestions regarding the improvement of our manuscript.

To verify that blood cells were able to survive the sharp ends of the nanowires, we used calcein-AM to determine the cell viability. The experimental details were add on line 7 of page 12.

As shown in Supplementary Fig. 17, the damage of sharp NWs for blood cells is negligible, which could be attributed to two reasons. First, the pore size of CF (~200 μm) is large enough to allow the blood cells smoothly through the dialyzer. Second, the NWs are coated with Con A that binds specifically to the surfaces of clinical pathogens, rather than blood cells (line 12 of page 6).

Comment 1: you have not reviewed the literature sufficiently - there are many groups who have successfully removed pathogens at higher flow rates (>100 mL/min) than you quote e.g. McCrea K et al (2014) Removal of Carbapenem-Resistant Enterobacteriaceae (CRE) from Blood by Heparin- Functional Hemoperfusion Media. PLoS ONE 9(12): e114242. Didar et al (2015) Improved treatment of systemic blood infections using antibiotics with extracorporeal opsonin hemoadsorption Biomaterials. 2015 Oct;67:382-92.

Reply: Agree, we cited them as ref 8 and 9 in revised manuscript. Although some groups have successfully removed the pathogens at high flow rates via the only force of synthetic small molecules, natural lectin, and antibodies, the major scientific challenge of this manuscript is to improve the captured efficiency. The shearing force of patient bloodstream results in the desorption of immobilized bacteria. To address this problem, we demonstrated that the significant bending of polycrystalline nanowires produced by weak carbohydrate-protein-binding forces could improve the bacterial capture efficiency from 40% on unbendable nanowires to 97% on bendable nanowire.

Comment 2: You have focused the paper on the nanowire technology, but in the blood experiments, you add ConA - which captures pathogens by lectin affinity, not by the 3D nanoclaws on which you base the paper.

Reply: Yes, the pathogens were captured from bloodstream by specific binding of ligand-receptor (Con A-mannose). The function of 3D nanoclaws is to improve significantly captured efficiency from flow bloodstream and avoid desorption of immobilized bacteria (Fig 5). In the unclosed state, the bendable polycrystalline nanowires are straight (pointing outwards); however, in the closed state, as long as the targeted bacteria are trapped in the nanowire arrays, these nanowires are instantly triggered, bending to close the trap. The formation of 3D nanoclaws must be triggered by driving force, which was generated by specific lectin-carbohydrate recognition between Con A and mannose of the bacterial surface lipopolysaccharide.

Comment 3: You do not give sufficient detail on the preservatives used to prevent blood clotting, you

have not tested for clotting or coagulation activation in running the septic blood through the filter, you have not made control filters.

Reply: Good suggestion. we forgot to give the experimental details about preventing blood clotting. The vacutainer with anticoagulants (heparin) coated was used to preserve the blood sample, which was added in method section (line 6 of page 12). In order to further illuminate the effect of coagulation, the parallel experiments were performed by a vacutainer with anticoagulants uncoated. As shown in Supplementary Fig. 15b, there is no significant difference in bacterial capture ability between anticoagulants coated and uncoated.

Comment 4: your claims are too strong, based on the single pathogen you have spiked into blood - the blood itself will clear > 90% of the pathogen by phagocytosis (depending on the preservative used and the temperature)

Reply: I agree the blood itself will clear > 90% of the pathogen by phagocytosis. The purpose of this manuscript is to demonstrate a conceptually new strategy for clearing pathogen that might not be killed by phagocytosis or when the concentration of bacteria in septic blood is higher than the clear ability of phagocytosis. As shown in Fig. 5e-h, the number of residual bacteria for smooth CF dialyzer were highest among the three dialyzers, which means phagocytosis failed to clean septic blood. In contrast, the ~97% bacteria were removed by the bendable NWS dialyzer.

To Reviewer #3 :

General comment: This paper describes new removal method of bacteria in bloodstream. Bendable nanowires modified with Con A indicates high bacterial capture efficiency. Especially reviewer interested in high capture efficiency by appropriate nanowire with Young's moduli. The results are clear and explain (discussion) is enough. This paper will be interested in the wide field of medical field or material science and micro/nano material science. And the concept will influence micro/nano material design. Minor comments:

Reply: We sincerely appreciate to this reviewer for his/her recommendations and constructive suggestions regarding the improvement of our manuscript.

Comment 1: P4, lines 137, "Such NW deformation has a negative effect on the bacteria-capturing capacity. The bacteria are only absorbed on the surfaces of bending nanostructures¹⁵, resulting in a low bacteria capturing efficiency." These sentences are important for topic (bendable nanowires) of this paper.

However, reviewer could not understand very well (immediately). Please insert below words in text because easy to understand. Or please explain more detail in text.

The bacteria are only absorbed on the surfaces of bending nanostructures, "and thus the bacteria will fall off readily". (although already written in Figure 3a caption)

Reply: It is very helpful. We inserted the sentence on line 9 of page 4 as your suggestion.

Comment 2: SEM images of the surface single-crystalline NWs/CF, polycrystalline NWs/CF at Figure

5c or supplementary Figure 13d and 13e are too small. Reader will be interested in the images of the adsorbed bacteria or blood cells. Please add expansion images at supplementary Fig 13.

Reply: Agree, we provided the enlarged SEM images of Fig. 5c as Supplementary Fig. 14, and expanded Supplementary Figure 13c, 13d and 13e as Supplementary Fig. 16.

Comment 3: Reader cannot immediately understand what blue fluorescence indicate. Please add words ex.“(Blue: DAPI)” at Figure 5.

Reply: Agree, and we added the sentence "The bacteria were stained with DAPI shown in blue" at Fig. 5 caption.

Comment 4: Have you put the anticoagulants in blood? If use, please describe in material and method.

Reply: Yes, we used the vacutainer with anticoagulants (heparin) coated to preserve the blood sample, which was added in method section (line 6 of page 12). In order to further illuminate the effect of coagulation, the parallel experiments were performed by a vacutainer with anticoagulants uncoated. As shown in Supplementary Fig. 15b, there is no significant difference in bacterial capture ability between anticoagulants coated and uncoated

Reviewers' Comments:

Reviewer #1:

Remarks to the Author:

The authors have addressed my comments. I agree with its publication after the authors make following adjustments.

Line 71-72: Blood is NOT entirely bacteria-free, it has been recently found to also host a microbiome that is not necessarily hostile to the host. Such as here

<https://www.ncbi.nlm.nih.gov/pubmed/26865079>

I recommend that they use 'pathogenic bacteria' in places appropriate when they would like to indicate the ones that we want to eradicate. This is also a good piece of knowledge for authors' future work, keeping in mind that we still have limited understanding on the functions of bacteria in our bloodstream and killing all bacteria without selectivity may not be a good thing.

The authors have not comprehended my comment on polysaccharides precisely. They are known as antigens, just that they do not have to be secreted into the blood stream. 'They secrete toxins...' can be revised into 'they produce antigens', so the authors do not have to go into details that may not be accurate.

Reviewer #2:

Remarks to the Author:

The authors have addressed my original concerns and the manuscript is acceptable for publication.

I suggest that Nature work with the authors to improve the english

To Reviewer #1:

General comment: The authors have addressed my comments. I agree with its publication after the authors make following adjustments.

Reply: We sincerely appreciate this reviewer for his/her praising comments, recommendations, and detailed suggestions regarding the improvement of our manuscript.

Comment 1: Line 71-72: Blood is NOT entirely bacteria-free, it has been recently found to also host a microbiome that is not necessarily hostile to the host. Such as here <https://www.ncbi.nlm.nih.gov/pubmed/26865079>

I recommend that they use 'pathogenic bacteria' in places appropriate when they would like to indicate the ones that we want to irradiate. This is also a good piece of knowledge for authors' future work, keeping in mind that we still have limited understanding on the functions of bacteria in our bloodstream and killing all bacteria without selectivity may not be a good thing.

Reply: We greatly appreciate the suggestion for using 'pathogenic bacteria' to make the expression of our manuscript more accurate, which were highlighted by yellow on line 30 of page 2.

Comment 2: The authors have not comprehended my comment on polysacchorides precisely. They are known as antigens, just that they do not have to be secreted into the blood stream. 'They secrete toxins...' can be revised into 'they produce antigens', so the authors do not have to go into details that may not be accurate.

Reply: Agree, the 'They begin secreting toxins' was replaced by 'They begin producing antigens' on line 31 of page 2.

To Reviewer #2:

General comment: The authors have addressed my original concerns and the manuscript is acceptable for publication. I suggest that Nature work with the authors to improve the English.

Reply: We sincerely appreciate to this reviewer for his/her positive affirmation. As for the English, this manuscript has been checked by at least two professional editors, both native speakers of English. For a certificate, please see: <http://www.textcheck.com/certificate/4qFRFRt>